# Beyond Weight-Only: Mixed-Precision Quantization for BERT Weights, Activations and Embeddings

## Abstract

Pre-trained language models deliver strong performance across various Natural Language Processing (NLP) tasks but remain costly to deploy due to memory and compute demands. To address this, model compression techniques such as pruning, knowledge distillation, and quantization have emerged, with quantization gaining traction due to hardware support for low precision. While uniform and extremely low-precision quantization have shown promise, mixed-precision approaches that assign variable bit-widths to weights/activations across the model offer a superior balance between compression and accuracy. In this work, we aim to evaluate the impact of mixed-precision quantization for inference on BERT language model. Unlike prior work that often neglects activation quantization, our study systematically explores both weights and activations in mixed-precision configurations. To further improve performance, we integrate knowledge distillation into the mixed-precision pipeline. We also evaluate the impact of quantization on the embedding layer, which is generally restricted solely to quantizing token weights. Evaluated on SQuAD and GLUE benchmarks, our results achieve substantial memory and computational reductions without sacrificing accuracy.

## 1 Introduction

Pre-trained language models such as BERT (Devlin et al., 2019) have achieved significant improvements across a wide range of NLP tasks (Wang, 2018; Rajpurkar, 2016), but these major gains come at the cost of massive parameter counts and high compute. For example, BERT$_{\text{BASE}}$ has 109M parameters (418MB memory size) and requires billions of operations. The need for real-time inference, along with privacy and security concerns, has made the efficient deployment of these models on edge devices a major challenge. Due to the limited resources, including energy, memory, and computational power of these devices, model compression is essential for their implementation. These concerns are also being addressed in data centers to reduce the energy consumption of these models. Various model compression methods have been successfully applied to BERT, such as pruning (Gordon et al., 2020), knowledge distillation (Sanh et al., 2019), and quantization (Zafrir et al., 2019). With the emergence of hardware platforms that offer better support for low (*e.g.* recent Nvidia GPUs and Google TPUs) and custom (*e.g.* FPGA and ASIC solutions) precision, quantization has become a particularly practical approach for efficient language model inference.

Quantization reduces model size and compute time by representing weights and activations at lower bit-widths, even changing the arithmetic. Transformer-based models can be uniformly quantized to 8 bits integer (Zafrir et al., 2019), and even in ternary (Zhang et al., 2020) or binary (Bai et al., 2020) representations with limited accuracy loss. Various methods (Wang et al., 2019; Dong et al., 2019; Huang et al., 2022; Yang & Jin, 2021) push the compression limit even further, using different bit-widths at the (sub)layer level. However, it is challenging to find efficient mixed-precision configurations that compress a model with minimal impact on accuracy. There are three main families of methods that attempt to optimize bit-width allocation for model compression: *search*, *metric* and *optimization*-based. Search-based methods iteratively explore the bit-width assignment space and are generally costly to use. Metric-based approaches are much faster, but tend to give sub-optimal results. Optimization methods offer good performance at a reasonable computational cost. Existing

mixed-precision model quantization approaches have mostly focused on weight quantization, since linear layers require the most memory.

In this paper, we push mixed-precision quantization beyond weights-only to include activations and all embedding parameters in BERT using AdaQAT (BlindCitation, 2024). AdaQAT is a gradient-based optimization framework for mixed-precision quantization that learns bit-widths to jointly quantize weights and activations during training, utilizing a Quantization-Aware Training (QAT) strategy. Its key lies in the use of relaxed fractional bit-widths, which are continuously optimized via gradient descent but discretized during both forward and backward passes. We first extend the method to per-layer mixed-precision optimization and then adapt AdaQAT to Large Language Models (LLMs) and integrate knowledge distillation to better preserve accuracy. We evaluate our approach on the SQuAD dataset, enabling direct comparison with state-of-the-art methods, and on GLUE benchmark, to investigate the impact of mixed-precision quantization over different language tasks. Our experiments demonstrate that the proposed method achieves substantial reductions in memory and computational overhead for $BERT_{BASE}$ while maintaining task accuracy. In addition, we examine the effect of quantizing the embedding layer's weights on model accuracy. While conventional approaches often focus exclusively on token embeddings, assuming key and position weights are negligible, our findings demonstrate that all embedding weights can be quantized with minimal impact on performance. For instance, on the SQuAD dataset, quantizing the full embedding layer to 4 bits or below preserves accuracy while halving the memory requirements of the embedding layer. The tool and experiments are open-source and available[1].

Our contributions are threefold:

- Extension of mixed-precision quantization to weights, activations, and embeddings in BERT via AdaQAT with integrated knowledge distillation.
- First systematic study of full embedding quantization (token, positional, and key embeddings), demonstrating negligible accuracy loss at low bit-widths.
- State-of-the-art accuracy–efficiency tradeoffs on SQuAD and GLUE, with substantial memory and compute reductions.

This article is structured as follows. Section 2 reviews related work on quantization-aware training and mixed-precision methods. Section 3 describes our methodology, including the integration of knowledge distillation into the mixed-precision flow. Finally, Section 4 presents experimental results, analyzing the performance of mixed-precision quantization applied to the $BERT_{BASE}$ model and the impact of embedding layer quantization.

## 2 RELATED WORK

Quantization can be applied either during or after training, leading to two primary approaches in practice. The first, Post-Training Quantization (PTQ), applies quantization after training and is fast to deploy, but can incur non-negligible accuracy loss at very low bit-widths (Banner et al., 2019). In contrast, Quantization-Aware Training (QAT), while slower and requiring access to training data, generally leads to better results.

**Quantization-Aware Training (QAT)** This method has been widely studied to train CNNs with small bit-widths. QAT methods stem from pioneering work on binary neural networks (Courbariaux et al., 2015; 2016). At its core, a QAT method consists of using a quantized version of the network during training in both forward and backward passes, while performing updates on full-precision copies of the network parameters. These full-precision parameters are then quantized to be used in the next iteration. A crucial aspect is how to perform backpropagation through quantized variables (parameters and activations). In the binary case, this was done using a so-called Straight-Through Estimator (STE) (Bengio et al., 2013) and this approach was later extended to cover larger bit-widths (Zhou et al., 2016), while also applying quantization to gradient signals. To further improve the accuracy of quantized networks, the STE idea can also be used to learn the parameters of uniform quantizers, such as scaling factors and bias terms for weight quantization (Esser et al., 2019; Bhalgat et al., 2020), and in the case of ReLU-based activations, clipping parameters (Choi et al., 2018).

---

[1]The anonymous code is available at `https://anonymous.4open.science/r/AdaQAT-A605/`

QAT has been successfully applied to LLMs, with BERT being trained in 8-bit integer QAT with minimal accuracy loss using STE-based approaches (Zafrir et al., 2019). Quantization with lower bit-widths has also been explored, including ternary (Zhang et al., 2020) and binary (Bai et al., 2020; Qin et al., 2022) quantized BERT. Tang et al. (2022) proposed to replace the STE-based gradient for updating the quantization scale with an MSE-based gradient strategy. The SQuAT (Wang et al., 2022) method alternates training between the sharpness objective and the step-size objective, allowing the model to learn the most appropriate parameter update magnitude to reach convergence close to flat minima.

**Mixed-Precision for Inference**   Mixed-precision quantization aims to reduce bit-widths to a finer grain, generally to a (sub)layer level, which can be tricky without an automatic approach to determine a correct configuration. This problem has been addressed using various approaches.

Search-based methods like HAQ (Wang et al., 2019) rely on reinforcement learning with hardware (latency & energy) feedback in the agent, whereas neural architecture search works like DNAS (Wu et al., 2018) uses gradient-based information. The major downside of using them is that they require significant time and computational resources. AQ-BERT (Zhao et al., 2021) adopts a Network Architecture Search approach to set different precision for each neuron sub-group of BERT.

Much faster results can be obtained using metric-based methods. For example, HAWQ (Dong et al., 2019) uses Hessian spectrum information at each layer to assign precisions; this approach was used in Q-BERT (Shen et al., 2020). Methods like in Yao et al. (2021); Ma et al. (2021) rely on linear programming models, while Liu et al. (2021) encourage quantization that leads to reduced sharpness in the task loss function. These approaches are fast but can be sub-optimal compared to more global optimization strategies (Huang et al., 2022).

Optimization-based approaches formulate the bit-width assignment as an optimization problem, with the main challenge being how to handle the fact that the loss is non-differentiable w.r.t. the bit-widths. Methods like FracBits (Yang & Jin, 2021) and BitPruning (Nikolić et al., 2020) use fractional bit-widths and linear interpolation during the forward path, whereas SDQ (Huang et al., 2022) is based on stochastic quantization, but seems limited to weight quantization. These methods work well in fine-tuning scenarios, but are unstable or do not work when training from scratch.

AdaQAT belongs to the optimization-based family and offers a flexible QAT framework that learns relaxed (fractional) bit-widths via gradient-based optimization. This paper extends AdaQAT to LLMs and improves its applicability to BERT, including joint treatment of activations and embeddings.

**Knowledge Distillation**   Knowledge distillation (KD) is a compression method that mimics the behavior of a large model, called the teacher, into a smaller lightweight model, called the student, by transferring learned knowledge. The concept of knowledge distillation (Buciluă et al., 2006) was first introduced as a method for transferring information from a large model to train a small model. It can be applied at various points in the network, such as for activations (Heo et al., 2019), neurons (Huang & Wang, 2017), or features of intermediate layers (Romero et al., 2014). KD was later used in semi-supervised learning to transfer information between a fully supervised teacher and student models using unlabeled data (Urner et al., 2011). This method was later augmented with the student model being trained to imitate the class probabilities of the softmax output of the teacher model (Hinton et al., 2015). KD has been successfully applied to smaller BERT-based models such as TinyBERT (Jiao et al., 2019) or DistilBERT (Sanh et al., 2019). This approach can be combined with quantization to reduce the effects of decreasing the precision (Kim et al., 2019; Shin et al., 2020; Boo et al., 2021; Tang et al., 2022; Kim et al., 2022) and works well even for extreme quantization in TernaryBERT (Zhang et al., 2020) and BinaryBERT (Bai et al., 2020).

## 3   MIXED-PRECISION EXPLORATION WITH ADAQAT

For simplicity, we start by presenting AdaQAT in the context of learning only one bit-width for all weights in the network and similarly for activations. We then extend the method to per-layer bit-width allocation.

## 3.1 OBJECTIVE FUNCTION

To learn the bit-widths of the uniform quantizers for both weights and activations, we use two real-valued variables $N_{\mathbf{w}}$ and $N_{\mathbf{a}}$, respectively. The actual integer bit-widths of the quantized network will be $\lceil N_{\mathbf{w}} \rceil$ and $\lceil N_{\mathbf{a}} \rceil$.

We use a regularization approach to model the loss function that takes into account the cost of a particular bit-width configuration, which corresponds to:

$$\mathcal{L}_{\text{Total}} = \mathcal{L}_{\text{Task}} \left( \mathbf{Y}_{\text{U,U}}^{(L)}; \mathbf{T} \right) + \lambda \mathcal{L}_{\text{Hard}} \left( \lceil N_{\mathbf{w}} \rceil, \lceil N_{\mathbf{a}} \rceil \right), \tag{1}$$

where $\lambda > 0$ is a regularization hyper-parameter between the main task loss $\mathcal{L}_{\text{Task}}$ (*i.e.,* the usual loss for the task that needs to be addressed) and the regularizing hardware loss $\mathcal{L}_{\text{Hard}}$ that measures the complexity of the model (in this case in terms of hardware cost). Here, $\mathbf{Y}_{\text{U,U}}^{(L)}$ represents the output of the $L$-layer quantized network we want to optimize, where its weights have been quantized to $\lceil N_{\mathbf{w}} \rceil$ bits (the first U in the superscript notation) and its activations to $\lceil N_{\mathbf{a}} \rceil$ bits (the second U in the superscript) on a current minibatch of data and $\mathbf{T}$ is the expected output.

FracBits (Yang & Jin, 2021) has reviewed various methods used to model the hardware cost of arithmetic precision choices for weights and activations. They argue in favor of memory size if only targeting weight quantization, and BitOPs for joint weight and activation quantization. As an example, for a linear layer, the BitOPs metric corresponds to

$$\text{BitOPs}(l) = \lceil N_{\mathbf{w}} \rceil \lceil N_{\mathbf{a}} \rceil |w_l|, \tag{2}$$

where $l$ is a linear layer and $|w_l|$ denotes the number of weights in the linear layer $l$. Thus, the overall BitOPs of a model corresponds to the sum of each linear layer's BitOPs.

In our particular case, since we are using one bit-width per weights and one per activations, the overall BitOPs hardware cost will be linear in $\lceil N_{\mathbf{w}} \rceil \lceil N_{\mathbf{a}} \rceil$, and we can simplify the regularization term to

$$\mathcal{L}_{\text{Hard}} \left( \lceil N_{\mathbf{w}} \rceil, \lceil N_{\mathbf{a}} \rceil \right) = \lceil N_{\mathbf{w}} \rceil \lceil N_{\mathbf{a}} \rceil. \tag{3}$$

Using Equation 3 will impact the value we need to consider for $\lambda$, which will be smaller than when integrating the scaling factors specific to each layer (determined by the layer type and shape) directly into $\mathcal{L}_{\text{Hard}}$.

## 3.2 BIT-WIDTH GRADIENTS AND PARAMETER UPDATES

Since the task loss $\mathcal{L}_{\text{Task}}$ is not directly differentiable with respect to the bit-width parameters, we use finite difference approximations as

$$\begin{aligned}
\frac{\partial \mathcal{L}_{\text{Task}}}{\partial N_{\mathbf{w}}} &\approx \mathcal{L}_{\text{Task}}(\mathbf{Y}_{\text{U,U}}^{(L)}; \mathbf{T}) - \mathcal{L}_{\text{Task}}(\mathbf{Y}_{\text{D,U}}^{(L)}; \mathbf{T}), \\
\frac{\partial \mathcal{L}_{\text{Task}}}{\partial N_{\mathbf{a}}} &\approx \mathcal{L}_{\text{Task}}(\mathbf{Y}_{\text{U,U}}^{(L)}; \mathbf{T}) - \mathcal{L}_{\text{Task}}(\mathbf{Y}_{\text{U,D}}^{(L)}; \mathbf{T}).
\end{aligned} \tag{4}$$

The U/D subscripts in the $\mathbf{Y}^{(L)}$ network outputs denote how the weights (first superscript) and activations (second superscript) have been quantized: U represents $\lceil N_{\mathbf{w}} \rceil$ or $\lceil N_{\mathbf{a}} \rceil$ and D represents $\lfloor N_{\mathbf{w}} \rfloor$ or $\lfloor N_{\mathbf{a}} \rfloor$.

The gradients of the total loss Equation 1 w.r.t. the bit-widths are then approximated as

$$\frac{\partial \mathcal{L}_{\text{Total}}}{\partial N_{\mathbf{x}}} \approx \frac{\partial \mathcal{L}_{\text{Task}}}{\partial N_{\mathbf{x}}} + \lambda \frac{\partial \mathcal{L}_{\text{Hard}}}{\partial \lceil N_{\mathbf{x}} \rceil}, \tag{5}$$

which are then used to update the fractional bit-width parameters. The gradient descent rule that does this takes the form

$$N_{\mathbf{x}}^{+} = N_{\mathbf{x}} - \eta_{\mathbf{x}} \frac{\partial \mathcal{L}_{\text{Total}}}{\partial N_{\mathbf{x}}}, \tag{6}$$

with $\mathbf{x} \in \{\mathbf{w}, \mathbf{a}\}$, $N_{\mathbf{x}}^{+}$ the new bit-width values at the next iteration, and $\eta_{\mathbf{x}} > 0$ the corresponding learning rates.

To summarize, during the *forward propagation phase* (forward pass computations through each layer are done using the `forward` function), the weights and activations are quantized according to the learned integer bit-widths, $\lceil N_{\mathbf{w}} \rceil$ and $\lceil N_{\mathbf{a}} \rceil$, respectively, and to the quantization policy. In the *gradient estimation phase*, the bit-width gradients are approximated according to Equation 4 and Equation 5, while in the *parameter update phase*, we use Equation 6 to update the bit-width parameters. The rest of the network (*e.g.,* weights and bias terms) and quantizer parameters are updated in parallel using standard optimizers, with their separate hyperparameters. The detailed algorithm is given in the appendix A.2.

**Convergence Behavior**   When $N_{\mathbf{w}}$ and $N_{\mathbf{a}}$ reach their optimized values, continuing to decrease them will lead to a (steep) increase of the task loss $\mathcal{L}_{\text{Task}}$ and consequently of $\mathcal{L}_{\text{Total}}$. This means that their gradient estimates in Equation 5 will become negative, and the gradient descent rule in Equation 6 will start increasing $N_{\mathbf{w}}$ and $N_{\mathbf{a}}$. Then, an oscillatory pattern forms (an example is provided in (BlindCitation, 2024, Fig. 1)). This oscillation phenomenon is not new, being present in several STE-based gradient approximation methods (Nagel et al., 2022; Kuzmin et al., 2022; Liu et al., 2023). When this happens, we monitor the number of oscillations, and as soon as it passes a certain threshold (which we empirically set to 10), we fix the bit-widths to $\lceil N_{\mathbf{w}} \rceil$ and $\lceil N_{\mathbf{a}} \rceil$, respectively, and continue the rest of the quantization process in standard QAT fashion.

## 3.3   Per-layer Extension

Extending the method to layer-wise precision configurations requires finding appropriate bit-width variables for each layer of the model such that, for an $L$-layer model, $\left\lceil N_{\mathbf{w}}^{(i)} \right\rceil$ and $\left\lceil N_{\mathbf{a}}^{(i)} \right\rceil$ are the bit-widths for layer $i \in [\![1..L]\!]$. This problem is non-trivial, since the computational cost of the gradient estimation phase would grow at least linearly with the number of bit-widths to learn. Instead, it is more cost-effective to focus on updating the bit-width of only one layer's weights or activations at a time, potentially switching the layer to update at each iteration. The overall approach can thus be viewed as a quantized version of a *block coordinate descent* (Tseng, 2001) update rule. Block coordinate optimization methods update only a group of parameters at a time and have emerged as viable approaches to solve complex nonconvex optimization problems, like those involving "big data" (see for instance Xu & Yin (2017) and the references therein).

Careful attention must be paid when selecting which bit-widths to update at a given iteration. The most straightforward approach is to take a bit-width parameter at random, but this turns out to be highly suboptimal in our case. Greedy-type approaches offer better results: at each iteration, update (*e.g.,* decrease) the bit-width that are most likely to degrade accuracy the least on the current training batch.

**Weights and Activations**   Similarly to the per-network bit-width learning context, one must decide if weight and activation bit-widths should be updated simultaneously or in sequence. Whereas in the per-network setting model accuracy is not sensitive to this choice (simultaneous and in-sequence updates tend to behave analogously), at finer granularities (*e.g.,* layer or channel) this is not the case. Taking bit-widths separately allows more freedom in exploring the search space. It is also generally the case that activations are more sensitive to perturbations than weights, meaning that model accuracy is less impacted by focusing on weights bit-width learning first and then moving on to activation bit-widths. The fact that activation values depend on the weights (and inputs) also makes it more natural to start with weights.

**Bit-width update selection metrics**   Choosing to always update the bit-width that leads to the least significant reduction (*i.e.,* with the smallest cost) is a sensible approach. As so, we adopted a Mean Squared Error (MSE) strategy for selecting which bit-width to update in the subsequent block coordinate descent iteration. For two quantized tensors of the same shape $\widetilde{\mathbf{X}}_1$ and $\widetilde{\mathbf{X}}_2$ (*i.e.,* either weights or activations, before and after the corresponding bit-width parameter has been updated), the cost is their mean squared error:

$$\text{MSE}\left(\widetilde{\mathbf{X}}_1, \widetilde{\mathbf{X}}_2\right) := \frac{1}{\text{size}(\widetilde{\mathbf{X}}_1)} \left\| \widetilde{\mathbf{X}}_1 - \widetilde{\mathbf{X}}_2 \right\|_F^2, \tag{7}$$

where $\text{size}(\widetilde{\mathbf{X}}_1)$ is the number of elements in $\widetilde{\mathbf{X}}_1$ and $\| \cdot \|_F$ is the Frobenius norm.

### 3.3.1 INCLUDING KNOWLEDGE DISTILLATION

Knowledge distillation combined with QAT has proven to be an effective combination for improving quantized model accuracy, even in mixed-precision contexts (Zhao et al., 2021). In this work, we do not claim to improve existing techniques, but rather propose adding knowledge distillation to the AdaQAT pipeline.

AdaQAT can be seen as a composite of two distinct stages. First, a mixed-precision exploration stage, followed by a standard QAT stage. This second stage fine-tunes the model using the mixed-precision configuration determined in the first stage. To prevent interference during exploration, knowledge distillation is added only during the refinement stage.

In this work, we use the knowledge distillation approach that has been successfully used to train TernaryBERT (Zhang et al., 2020). In this scenario, the quantized BERT acts as the student model, and learns to recover the behaviours of the full-precision teacher model over the Transformer layers and prediction layer. The distillation objective for the Transformer layers combines two parts. First, a knowledge distillation loss that distills embeddings and the outputs of all Transformer layers from the full-precision teacher model to the quantized student model. Second, an attention-based distillation loss that aligns the student's attention scores, across all heads in each Transformer layer with those of the teacher model. Thus the distillation for a Transformer layers $\mathcal{L}_{\text{trm}}$ is formulated as:

$$\mathcal{L}_{\text{trm}} = \sum_{l=1}^{L+1} \text{MSE}\left(\mathbf{H}_l^S, \mathbf{H}_l^T\right) + \sum_{l=1}^{L} \text{MSE}\left(\mathbf{A}_l^S, \mathbf{A}_l^T\right). \tag{8}$$

In addition to the Transformer layers, knowledge is distilled to the prediction layer, where the student model's logits $\mathbf{P}^S$ learn to fit the teacher model's logits $\mathbf{P}^T$ using a soft cross-entropy loss:

$$\mathcal{L}_{\text{pred}} = \text{SCE}\left(\mathbf{P}^S, \mathbf{P}^T\right). \tag{9}$$

All leading to the following global objective of knowledge distillation in the training process:

$$\mathcal{L} = \mathcal{L}_{\text{trm}} + \mathcal{L}_{\text{pred}}. \tag{10}$$

## 4 EXPERIMENTS AND DISCUSSION

To evaluate the performance of mixed-precision models and the impact of embedding quantization on accuracy, we perform several mixed-precision quantization experiments on SQuAD datasets (Rajpurkar, 2016; Rajpurkar et al., 2018) and the GLUE benchmark (Wang, 2018), and compare the results with those obtained with the state-of-the-art.

We measure the efficiency of the models via memory size and the BitOPs metric (see equation 2). As the state-of-the-art mixed-precision models do not give a precise information on bit-widths used on each layer, we approximate the BitOPs value using the memory size. Since activations are uniform across all layers, it is possible to factorize them to obtain a sum that depends only on the weights and their bit-width, which results in calculating the memory size of weights (excluding the embedding weights).

$$\text{BitOPs}(M) = N_{\mathbf{a}} \sum_{i=0}^{L-1} \left\lceil N_{\mathbf{w}}^{(i)} \right\rceil |w_l| = N_{\mathbf{a}} \times \text{memsize}(W) = N_{\mathbf{a}} \times \text{Size}(W) \times 8 \cdot 10^6$$

where $M$ is an $L$-layer model and memsize$(W)$ is computed in MB.

All tables report the bit-width configurations in the W-E-A format, which corresponds to weights of linear layers, word embedding and activation layers, respectively. As state-of-the-art methods do not give precise information on mixed-precision configuration, we put MP to indicate variable-length layers. The Size includes the embedding weights, if not stated otherwise. Depending on the task, we compare the exact match (EM) and the $F_1$-score.

**Quantization Procedures** All of our experiments were performed using PyTorch 1.13. We quantize the weights and the inputs (referred to as 'activations' in the following pages) from all Transformer layers. For a fair comparison with state-of-the-art results, only word embeddings are quan-

Table 1: Effect of adding knowledge distillation into the AdaQAT pipeline on quantized BERT$_{\text{BASE}}$ models accuracy evaluated on the SQuAD v2.0 dataset.

| | Model | Bit-width W-E-A | SQuAD v2.0 EM | SQuAD v2.0 F$_1$-score | BitOPs ($\times 10^9$) |
|---|---|---|---|---|---|
| FP baseline | BERT$_{\text{BASE}}$ | FP32 | 74.5 | 77.7 | / |
| AdaQAT without Distillation | Uniform | 2-4-4 | 73.7 | 77.0 | 0.684 |
| | Mixed-precision | 2.08-4-3.93 | 74.0 | 77.2 | 0.677 |
| AdaQAT with Distillation | Uniform | 2-4-4 | 75.0 | 77.9 | 0.684 |
| | Mixed-precision | 2.08-4-3.93 | 75.2 | 77.9 | 0.677 |

Table 2: Comparison with state-of-the-art quantization methods on SQuAD datasets.

| Model | Bit-width W-E-A | Size (MB) | BitOPs ($\times 10^9$) | SQuAD v1.1 EM | SQuAD v1.1 F$_1$-score | SQuAD v2.0 EM | SQuAD v2.0 F$_1$-score |
|---|---|---|---|---|---|---|---|
| BERT$_{\text{BASE}}$ | FP32 | 417.6 | / | 81.5 | 88.7 | 74.5 | 77.7 |
| TernaryBERT | 2-2-8 | 28.0 | 1.368 | 80.1 | 87.5 | 73.3 | 76.6 |
| BinaryBERT | 1-1-4 | 16.5 | 0.342 | 79.3 | 87.2 | 72.5 | 75.4 |
| Q-BERT | MP-8-8 | 53.2 | $\approx 1.80$ | 79.85 | 87.49 | / | / |
| AQ-BERT | MP-8-8 | 55.0 | $\approx 1.92$ | 79.85 | 87.00 | / | / |
| AdaQAT | 2.08-4-3.93 | 29.2 | 0.677 | / | / | 75.2 | 77.9 |
| | 2.06-4-3.96 | 35.9 | 0.701 | 81.07 | 88.71 | / | / |

tized in the sections on mixed-precision. The embedding bit-width is also not included in the mixed-precision search space. Instead, the embedding layer quantization is explored in detail in the experiments of the Section 4.4. In all experiments we do not quantize the bias in the linear layer, the softmax operation, layer normalization and the last task-specific layer. We use LSQ (Esser et al., 2019) as a quantization strategy to learn the ranges for both weights and activations. We use the full-precision BERT fine-tuned on the downstream task to initialize our quantized model. All our training settings and hyper-parameters are listed in Appendix A.1.

**SQuAD datasets**   is a reading comprehension dataset consisting of more than 100,000 crowd-sourced questions based on Wikipedia articles in its first version: SQuAD v1.1 (Rajpurkar, 2016). Each entry contains a context paragraph, a question, and the corresponding answer span. The dataset was expanded with SQuAD v2.0 (Rajpurkar et al., 2018), which introduced over 50,000 additional unanswerable questions.

**GLUE benchmark**   is a standardized collection of nine diverse natural language understanding tasks built on established existing datasets and selected to cover a diverse range of dataset sizes, and degrees of difficulty and text genres, including sentiment analysis, textual entailment, paraphrase detection, and linguistic acceptability (Wang, 2018). The benchmark is designed to evaluate the performance of models on a wide range of linguistic phenomena found in natural language.

### 4.1 BENEFITS OF KNOWLEDGE DISTILLATION

Knowledge distillation procedures are orthogonal to quantization and mixed-precision. In Table 1, we investigate the effect of incorporating knowledge distillation in the AdaQAT pipeline evaluated on the SQuAD v2.0 dataset with BERT$_{\text{BASE}}$. As observed in previous SotA work (Zhao et al., 2021), this combination enables the quantized model to increase its predictive performance. The accuracy of uniform and mixed-precision quantization BERT$_{\text{BASE}}$ models increased by more than $1\%$ for EM and $\approx 1\%$ for the F$_1$-score when knowledge distillation is added to the AdaQAT pipeline. Hence, we use knowledge distillation in all further experiments.

### 4.2 COMPARISON OF ADAQAT WITH SOTA

Table 2 shows the results of applying AdaQAT on SQuAD datasets compared to other methods from the literature. The first line shows the floating-point baseline result, whereas the second group of lines showcases *static* methods, where activations are not quantized and weights are quantized

Table 3: State-of-the-Art quantization results on the GLUE benchmark.

| Model | Bit-width W-E-A | Size (MB) | BitOPs ($\times 10^9$) | MNLI -m/mm | QQP | QNLI | SST-2 | CoLA | STS-B | MRPC | RTE |
|---|---|---|---|---|---|---|---|---|---|---|---|
| BERT$_{BASE}$ | FP32 | 417.6 | / | 84.57/84.46 | 91.39 | 91.34 | 92.32 | 59.67 | 90.14 | 86.28 | 69.93 |
| Q-BERT | 2-8-8 | 43.1 | 1.368 | 76.56/77.02 | / | / | 84.63 | / | / | / | / |
| | MP-8-8 | 53.2/53.2 | $\approx 1.80$ | 83.51/83.55 | / | / | 92.55 | / | / | / | / |
| | MP-8-8 | 46.1/48.1 | $\approx 1.35/1.48$ | 81.75/82.29 | / | / | 92.08 | / | / | / | / |
| TernaryBERT | 2-2-8 | 28.0 | 1.368 | 83.0/82.2 | 88.4 | 90.0 | 92.9 | 47.8 | 84.3 | 87.5 | 68.4 |
| BinaryBERT | 1-1-4 | 16.5 | 0.342 | 83.9/84.2 | 91.2 | 90.9 | 92.3 | 44.4 | 87.2 | 83.3 | 65.3 |

Table 4: Our mixed-precision results with AdaQAT on the GLUE benchmark.

| | MNLI -m/mm | QQP | QNLI | SST-2 | CoLA | STS-B | MRPC | RTE |
|---|---|---|---|---|---|---|---|---|
| Accuracy (%) | 84.66/84.38 | 90.96 | 91.72 | 92.43 | 58.02 | 89.73 | 86.50 | 69.10 |
| Bit-width W-E-A | 2.25-4-4.11 | 2.16-4-4.03 | 2.18-4-4.14 | 2.16-4-4.07 | 2.41-4-4.30 | 2.27-4-4.21 | 2.24-4-4.18 | 2.47-4-4.34 |
| Size (MB) | 36.67 | 36.23 | 36.75 | 37.11 | 39.77 | 37.70 | 37.04 | 40.28 |
| BitOPs ($\times 10^9$) | 0.756 | 0.737 | 0.773 | 0.781 | 0.902 | 0.815 | 0.792 | 0.925 |

uniformly. The BitOPs for Q-BERT and AQ-BERT were not provided by the authors, hence we approximate them using the given memory size and the model linear weights.

The third group of lines corresponds to mixed-precision methods where the weight bit-width is learned and the activations are quantized uniformly to a manually set bit-width. The last groups of lines present our mixed-precision BERT$_{BASE}$ results on SQuAD. EM and $F_1$-score are similar to or slightly better than the FP32 baseline. Compared to the other mixed-precision models Q-BERT and AQ-BERT, considering activations significantly reduces the computational cost of the model. Our configurations also significantly reduced the size required to store the weights in memory, while achieving better accuracy. Although mixed-precision models require more memory than binary or ternary models, they offer greater accuracy closer to the baseline. The computational cost of activations is also not negligible, with much higher BitOPs for TernaryBERT despite a smaller memory size for weights compared to our mixed-precision models.

### 4.3 MIXED-PRECISION FOR DIVERSE LANGUAGE TASKS WITH GLUE DATASET

Table 3 presents the state-of-the-art results for quantized BERT$_{BASE}$ models on the GLUE benchmark. Note that the mixed-precision approaches, up to our knowledge, have considered only a subset of GLUE datasets/tasks. Table 4 presents our mixed-precision results on the full GLUE benchmark.

Compared to Q-BERT, our approach demonstrates clearly superioir results in both accuracy and size. By including the embedding and activation layer quantization into the optimization objective, AdaQAT usually maintains the baseline accuracy drop within $1\%$ while dividing the memory footpring by roughly 11 compared to the FP32 baseline, and by 1.5 compared to Q-BERT. This demonstrates the robustness of BERT towards full model quantization.

Extremely low but uniform precision models, TernaryBERT and BinaryBERT, perform very well on most of the tasks but suffer on datasets that are sensitive to quantization, e.g. CoLa. With AdaQAT one can automatically find mixed-precision configurations that have slightly larger configurations that add the necessary bit-widths at appropriate layers. For instance, for CoLA we obtained that a confingration 2.41-4-4.30 limits the accuracy drop to only $1\%$, while the TernaryBERT 2-2-8 configuration suffers a roughly $12\%$ accuracy loss compared to the baseline. This non-trivial solution of decreasing activation bit-widths while non-homogeneously increasing weights and embeddings formats is impossible to find by hand but required an optimization-based approach.

### 4.4 QUANTIZATION EFFECTS ON THE EMBEDDING LAYER

The weights of the embedding layer account for 21% of the total parameters in the BERT$_{BASE}$ model. Of this 21%, 98% are token weights. For this reason, in SotA approaches, token weights are typically included in quantization exploration, while key and position weights are not quantized as they are considered negligible. While this is a reasonable argument, this perspective is inappropriate when considering compression for model deployment on resource-constrained devices.

Table 5: Evaluation of the quantization effects of embedding weights on various tasks. W-A represents the bit-width for weights and activations of linear layers, while T-K-P refers to token, key and position embedding, respectively. Each 32-bit bit-width corresponds to single precision. The specified memory size corresponds to the size of the embedding table. CR is the embedding weights compression ratio w.r.t. FP32 baseline.

| Model | Bit-width W-A | Embedding T-K-P | Embedding size (MB) | CR × | MNLI -m/mm | SST-2 | SQuAD v1.1 EM | SQuAD v1.1 $F_1$-score | SQuAD v2.0 EM | SQuAD v2.0 $F_1$-score |
|---|---|---|---|---|---|---|---|---|---|---|
| BERT$_{\text{BASE}}$ | FP32 | FP32 | 95.32 | 1.00 | 84.57/84.46 | 92.32 | 81.41 | 88.61 | 74.45 | 77.64 |
| Q-BERT | 8-8 | 8-32-8 | 23.84 | 4.00 | 83.83/83.91 | 92.88 | / | 88.47 | / | / |
| | | 4-32-8 | 12.12 | 7.86 | 82.91/83.67 | 91.74 | / | 87.55 | / | / |
| | | 8-32-4 | 23.64 | 4.03 | 82.84/82.25 | 89.11 | / | 72.38 | / | / |
| | | 4-32-4 | 11.92 | 8.00 | 78.08/78.96 | 85.55 | / | 61.70 | / | / |
| AdaQAT | 8-8 | 8-32-32 | 25.01 | 3.81 | 84.57/84.46 | 92.89 | 83.59 | 90.56 | 75.17 | 78.17 |
| | | 4-32-32 | 13.30 | 7.17 | 84.95/84.72 | 92.66 | 83.54 | 90.55 | 75.20 | 78.24 |
| | | 8-32-8 | 23.84 | 4.00 | 85.05/84.78 | 92.55 | 83.55 | 90.57 | 75.27 | 78.20 |
| | | 4-4-4 | 11.91 | 8.00 | 85.02/84.43 | 92.78 | 83.54 | 90.56 | 75.28 | 78.23 |
| | | 2-2-2 | 5.96 | 16.0 | 84.97/84.64 | 92.66 | 83.60 | 90.47 | 75.38 | 78.29 |
| | 2-4 | 4-32-32 | 13.30 | 7.17 | 83.89/84.14 | 91.17 | 81.58 | 89.00 | 74.98 | 76.86 |
| | | 4-32-4 | 11.92 | 8.00 | 83.76/83.60 | 91.17 | 81.47 | 89.00 | 74.80 | 77.60 |
| | | 4-4-4 | 11.91 | 8.00 | 83.78/84.11 | 91.17 | 81.51 | 89.00 | 74.43 | 77.44 |
| | | 2-2-2 | 5.96 | 16.0 | 83.79/84.30 | 90.60 | 81.36 | 88.91 | 74.11 | 77.14 |

Q-BERT (Shen et al., 2020) examines the impact of quantizing word embeddings and position embeddings in the BERT$_{\text{BASE}}$ model. The authors argue that the embedding layer is more sensitive to quantization than the linear layer, with performance drops of up to 10% on tasks like SST-2, MNLI, and CoNLL-03, and over 20% on SQuAD v1.1 when using 4-bit embedding quantization. Additionally, they find that position embeddings are more sensitive to quantization, leading to an extra 2% performance degradation compared to quantizing word embeddings alone. GOBO (Zadeh et al., 2020) observed different behaviors suggesting that it is nevertheless possible to lower the bit-width of the embeddings without significantly impacting accuracy, which we confirm with our experiments as well.

Table 5 shows the size of the embedding layer, model accuracy across multiple datasets and different quantization configurations. These results indicate that it is actually possible to quantize all weights in the embedding layer to 2-bit while maintaining the model accuracy drop below 2% on the evaluated datasets. The memory footprint of the embedding layer is reduced by a factor of 16 compared to the single-precision model and by a factor of 4.2 compared to the approach generally used in state-of-the-art work, i.e. quantizing only the token weights.

## 5 CONCLUSION & FUTURE WORK

In this paper, we improve the efficiency of BERT-based models by providing a mixed-precision algorithm and tool AdaQAT, which looks beyond weight layers and considers activation and embedding layers. Our new mixed-precision BERT$_{\text{BASE}}$ target low bitwidth ($< 8$) fixed-point representation across linear and activation layers and often maintain the baseline accuracy while reducing the computational cost by more than 2 and the memory size by 1.5, compared to the existing approaches that only take weight quantization into account. On top of that, we integrate knowledge distillation into the AdaQAT pipeline, increasing the accuracy by roughly 1%.

Furthermore, we evaluate the effect of quantizing all the weights (token, key, and position) of the embedding layer on model accuracy. Our results show that it is possible to quantize all weights in the embedding layer to 2 bits, while maintaining the model accuracy drop below 2% on the evaluated datasets. The memory footprint of the embedding layer is reduced by a factor of 16, compared to the FP32 model and by a factor of 4.2 compared to previous state-of-the-art work.

As future work, we will evaluate mixed-precision on smaller models such as TinyBERT (Jiao et al., 2019) or DistilBERT (Sanh et al., 2019), which are more sensitive to quantization than larger models such as BERT$_{\text{BASE}}$. We are also interested in evaluating this approach on domain-specific task models and datasets, which might be more sensitive to agressive quantization and where non-trivial mixed-precision assignments can find efficient configurations.

## 6 ETHICS STATEMENT

Compression and quantization can alter model behavior in certain ways, including that reduced precision may degrade accuracy, calibration, or robustness on under-represented inputs or demographic subgroups, potentially exacerbating fairness issues present in the teacher or training data. Knowledge distillation transfers model behaviors from teacher to student and, therefore, can propagate latent biases or failure modes. The models must therefore, as they are, be used with caution.

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

# A  APPENDIX

## A.1  TRAINING SETTINGS AND HYPER-PARAMETERS

Table 6: Hyper-parameters for training $\text{BERT}_{\text{BASE}}$ on SQuAD datasets and the GLUE benchmark

| | SQuAD v1.1 | v2.0 | MNLI -m/mm | QQP | QNLI | SST-2 | CoLA | STS-B | MRPC | RTE |
|---|---|---|---|---|---|---|---|---|---|---|
| Learning rate | | | | | $2 \times 10^{-5}$ | | | | | |
| LR decay | | | | | Linear | | | | | |
| Batch size | 16 | 16 | 32 | 32 | 32 | 32 | 16 | 32 | 32 | 32 |
| Max. seq. length | 384 | 384 | 128 | 128 | 128 | 64 | 64 | 128 | 128 | 128 |
| Eval step | 200 | 200 | 1000 | 1000 | 1000 | 200 | 50 | 50 | 200 | 100 |
| $\lambda_{\mathbf{w}}/\lambda_{\mathbf{a}}$ | | | | | 0.02/0.01 | | | | | |

## A.2  ADAQAT ALGORITHM

---

**Algorithm 1** AdaQAT iteration for updating the $N_{\mathbf{a}}, N_{\mathbf{w}}$ bit-widths of an $L$-layer model

---

**Require:** a minibatch of inputs $\mathbf{Y}^{(0)}$ and corresponding targets $\mathbf{T}$, weights $\mathbf{W} \in \mathbb{R}$, weight bit-width $N_{\mathbf{w}} \in \mathbb{R}_+^*$, activation bit-width $N_{\mathbf{a}} \in \mathbb{R}_+^*$, regularization parameter $\lambda > 0$, bit-width learning rates $\eta_{\mathbf{w}}, \eta_{\mathbf{a}} > 0$, loss functions $\mathcal{L}_{\text{Task}}$ and $\mathcal{L}_{\text{Hard}}$.

**Ensure:** updated parameters $N_{\mathbf{w}}^+$ and $N_{\mathbf{a}}^+$

**1. Forward propagation:**

1: $\widetilde{\mathbf{W}}_{\text{U}}^{(1)} \leftarrow \texttt{quantize\_w}\left(\mathbf{W}^{(1)}, 8\right)$

2: $\mathbf{Y}_{\text{U,U}}^{(1)}, \mathbf{Y}_{\text{D,U}}^{(1)}, \mathbf{Y}_{\text{U,D}}^{(1)} \leftarrow \texttt{forward}\left(\mathbf{Y}^{(0)}, \widetilde{\mathbf{W}}_{\text{U}}^{(1)}\right)$

3: **for** $k = 2$ **to** $L - 1$ **do**

4:      $\widetilde{\mathbf{W}}_{\text{U}}^{(k)} \leftarrow \texttt{quantize\_w}\left(\mathbf{W}^{(k)}, \lceil N_{\mathbf{w}} \rceil\right), \quad \widetilde{\mathbf{W}}_{\text{D}}^{(k)} \leftarrow \texttt{quantize\_w}\left(\mathbf{W}^{(k)}, \lfloor N_{\mathbf{w}} \rfloor\right)$

5:      $\mathbf{Y}_{\text{U,U}}^{(k)} \leftarrow \texttt{forward}\left(\texttt{quantize\_a}\left(\mathbf{Y}_{\text{U,U}}^{(k-1)}, \lceil N_{\mathbf{a}} \rceil\right), \widetilde{\mathbf{W}}_{\text{U}}^{(k)}\right)$

6:      $\mathbf{Y}_{\text{D,U}}^{(k)} \leftarrow \texttt{forward}\left(\texttt{quantize\_a}\left(\mathbf{Y}_{\text{D,U}}^{(k-1)}, \lceil N_{\mathbf{a}} \rceil\right), \widetilde{\mathbf{W}}_{\text{D}}^{(k)}\right)$

7:      $\mathbf{Y}_{\text{U,D}}^{(k)} \leftarrow \texttt{forward}\left(\texttt{quantize\_a}\left(\mathbf{Y}_{\text{U,D}}^{(k-1)}, \lfloor N_{\mathbf{a}} \rfloor\right), \widetilde{\mathbf{W}}_{\text{U}}^{(k)}\right)$

8: **end for**

9: $\widetilde{\mathbf{W}}_{\text{U}}^{(L)} \leftarrow \texttt{quantize\_w}\left(\mathbf{W}^{(L)}, 8\right)$

10: $\mathbf{Y}_{\text{U,U}}^{(L)} \leftarrow \texttt{forward}\left(\texttt{quantize\_a}\left(\mathbf{Y}_{\text{U,U}}^{(L-1)}, \lceil N_{\mathbf{a}} \rceil\right), \widetilde{\mathbf{W}}_{\text{U}}^{(L)}\right)$

11: $\mathbf{Y}_{\text{D,U}}^{(L)} \leftarrow \texttt{forward}\left(\texttt{quantize\_a}\left(\mathbf{Y}_{\text{D,U}}^{(L-1)}, \lceil N_{\mathbf{a}} \rceil\right), \widetilde{\mathbf{W}}_{\text{D}}^{(L)}\right)$

12: $\mathbf{Y}_{\text{U,D}}^{(L)} \leftarrow \texttt{forward}\left(\texttt{quantize\_a}\left(\mathbf{Y}_{\text{U,D}}^{(L-1)}, \lfloor N_{\mathbf{a}} \rfloor\right), \widetilde{\mathbf{W}}_{\text{U}}^{(L)}\right)$

**2. Gradient Estimation:**

13: $\dfrac{\partial \mathcal{L}_{\text{Task}}}{\partial N_{\mathbf{w}}} \approx \mathcal{L}_{\text{Task}}\left(\mathbf{Y}_{\text{U,U}}^{(L)}; \mathbf{T}\right) - \mathcal{L}_{\text{Task}}\left(\mathbf{Y}_{\text{D,U}}^{(L)}; \mathbf{T}\right)$

14: $\dfrac{\partial \mathcal{L}_{\text{Task}}}{\partial N_{\mathbf{a}}} \approx \mathcal{L}_{\text{Task}}\left(\mathbf{Y}_{\text{U,U}}^{(L)}; \mathbf{T}\right) - \mathcal{L}_{\text{Task}}\left(\mathbf{Y}_{\text{U,D}}^{(L)}; \mathbf{T}\right)$

15: $\dfrac{\partial \mathcal{L}_{\text{Total}}}{\partial N_{\mathbf{w}}} \approx \dfrac{\partial \mathcal{L}_{\text{Task}}}{\partial N_{\mathbf{w}}} + \lambda \dfrac{\partial \mathcal{L}_{\text{Hard}}}{\partial \lceil N_{\mathbf{w}} \rceil}$

16: $\dfrac{\partial \mathcal{L}_{\text{Total}}}{\partial N_{\mathbf{a}}} \approx \dfrac{\partial \mathcal{L}_{\text{Task}}}{\partial N_{\mathbf{a}}} + \lambda \dfrac{\partial \mathcal{L}_{\text{Hard}}}{\partial \lceil N_{\mathbf{a}} \rceil}$

**3. Parameter update:**

17: $N_{\mathbf{w}}^+ = N_{\mathbf{w}} - \eta_{\mathbf{w}} \dfrac{\partial \mathcal{L}_{\text{Total}}}{\partial N_{\mathbf{w}}}$

18: $N_{\mathbf{a}}^+ = N_{\mathbf{a}} - \eta_{\mathbf{a}} \dfrac{\partial \mathcal{L}_{\text{Total}}}{\partial N_{\mathbf{a}}}$

---

## A.3 MIXED-PRECISION CONFIGURATIONS

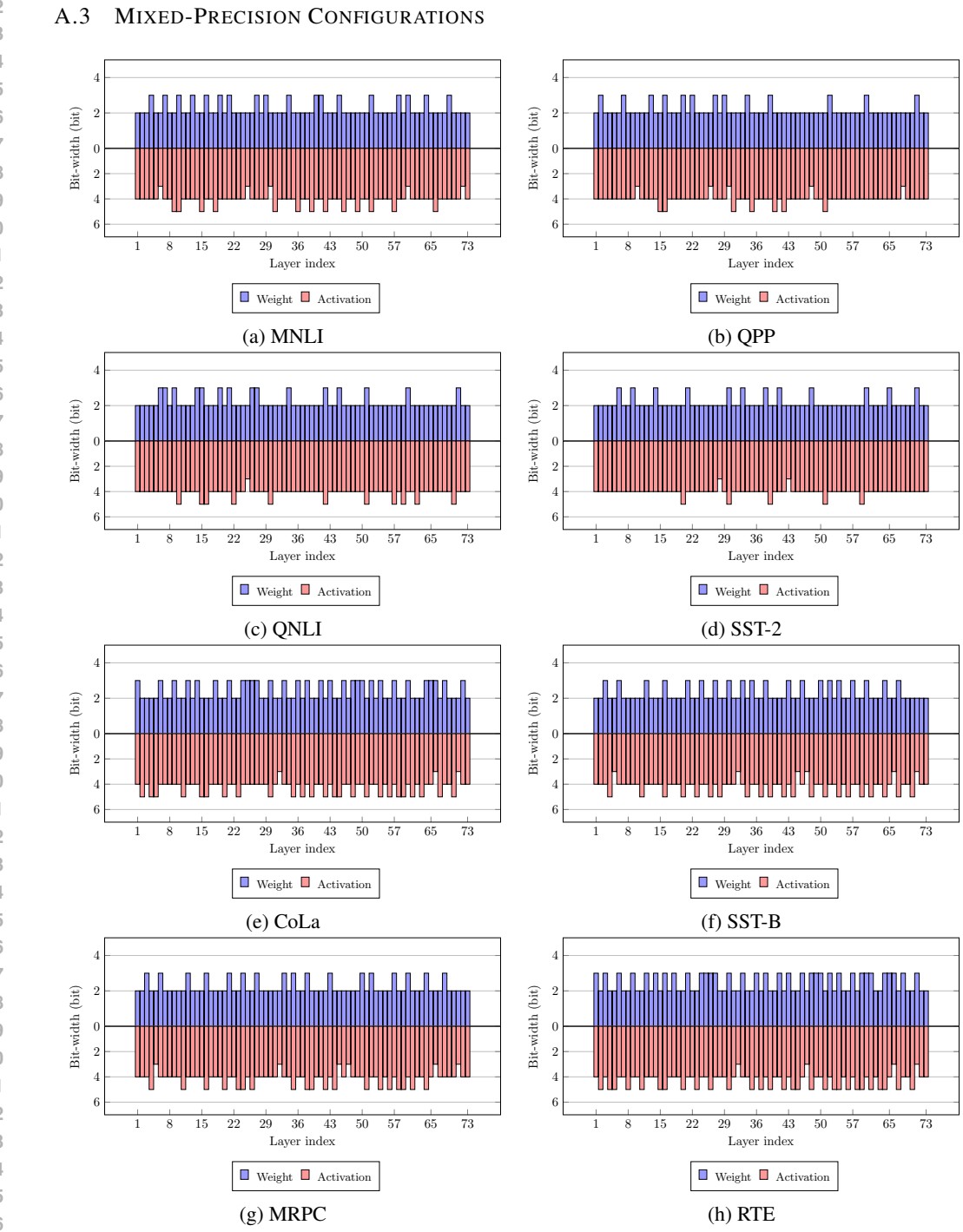

Figure 1: GLUE benchmark

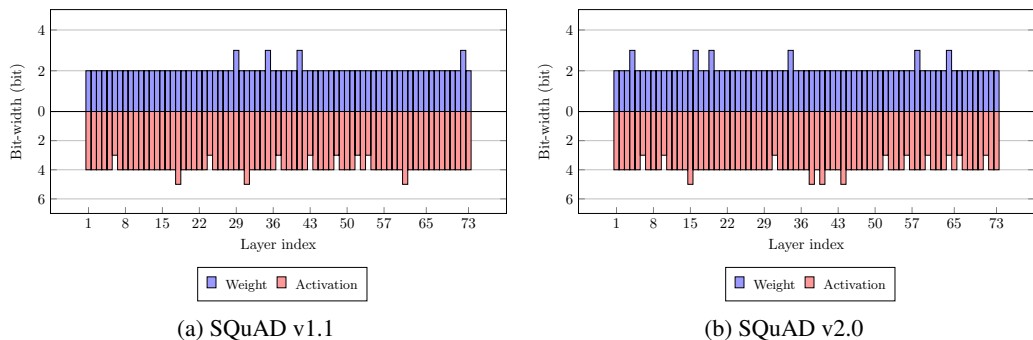

(a) SQuAD v1.1

(b) SQuAD v2.0

Figure 2: SQuAD datasets