# OpenReview forum: "Beyond Weight-Only: Mixed-Precision Quantization for BERT Weights, Activations and Embeddings"
_ICLR.cc/2026/Conference — ICLR 2026 Conference Withdrawn Submission_

### Official Review · Reviewer_QKGj · 2025-10-24

**Soundness:** 2
**Presentation:** 2
**Contribution:** 1
**Rating:** 2
**Confidence:** 4

**Summary:**

The paper extends mixed-precision quantization in BERT from weights-only to weights, activations, and all embedding parameters using AdaQAT, a gradient-based QAT framework that learns per-layer bit-widths via relaxed bit assignments. The method is adapted to LLMs (actually only BERT) and coupled with knowledge distillation to preserve accuracy. The proposed method achieves state-of-the-art accuracy–efficiency tradeoffs on SQuAD and GLUE, with substantial memory and compute reductions.

**Strengths:**

The paper is well written.

**Weaknesses:**

1. Clarification on “LLM” Definition

The authors state that they “adapt AdaQAT to Large Language Models (LLMs).” However, the experiments are conducted primarily on BERT models. In most contexts, “LLMs” typically refer to large-scale causal language models (e.g., GPT, LLaMA). It would be more precise to revise the wording or clarify whether the proposed method is also applicable to such models.

2. Motivation for Learnable bit-width in Activation Quantization

The paper extends AdaQAT by introducing learnable bit-widths for activations, yet the motivation behind this design is insufficiently explained. Unlike weight quantization, activation quantization is usually intended to accelerate inference by leveraging hardware-specific low-precision compute (e.g., NVIDIA GPUs support 1/4/8-bit tensor operations). The paper does not provide an analysis or empirical evidence regarding the practical benefits, e.g., latency or throughput improvements, of the proposed activation quantization scheme.

3. Outdated Baselines

The baseline methods (TernaryBERT, BinaryBERT, Q-BERT, AQ-BERT, all from 2020) are relatively outdated. To ensure a fair and convincing comparison, the authors should consider including more recent quantization methods for BERT or transformer models, which would better reflect the current state of the field.

**Questions:**

See Weaknesses

---

### Official Review · Reviewer_UWDy · 2025-11-01

**Soundness:** 2
**Presentation:** 2
**Contribution:** 1
**Rating:** 2
**Confidence:** 4

**Summary:**

The paper extends mixed-precision quantization of Transformer models beyond weights to include activations and embedding layers. Building on the AdaQAT framework, the authors integrate quantization-aware training (QAT) with knowledge distillation (KD) to learn layer-wise bit-widths for both weights and activations through gradient-based optimization. The method also quantizes all embedding components (token, positional, and key embeddings), a step often skipped in prior work. Evaluations on SQuAD v1.1/v2.0 and the GLUE benchmark show that the proposed approach can maintain accuracy close to full precision while reducing memory and computational cost by up to 1.5× over 8-bit baselines and over 10× versus FP32.

**Strengths:**

1. The paper provides a systematic extension of mixed-precision quantization to activations and embeddings, areas that are often neglected in existing quantization studies.

2. The integration of knowledge distillation within a gradient-based mixed-precision framework is executed cleanly and improves accuracy across tasks.

3.  The inclusion of embedding quantization ablation is a useful practical contribution, showing that 2-bit quantization of all embeddings can retain accuracy within 2 % of the FP32 baseline.

4. The writing is generally clear. Experiment details and tables are comprehensive, allowing replication of key results.

**Weaknesses:**

1. The proposed work mainly extends AdaQAT to a new domain (BERT/QAT with embeddings) without introducing a new algorithmic idea or optimization principle. The core fractional bit-width learning with finite-difference gradients mechanism is identical to existing mixed-precision schemes such as FracBits [1] and BitPruning [2]. As a result, the paper reads more like an engineering case study.

2. The approach remains empirically motivated. The choice of gradient approximations, oscillation thresholds, and per-layer update heuristics (e.g., MSE-based block selection in Section 3.3) lacks theoretical justification. The paper does not analyze why the method converges or why it should outperform simpler bit-width search or sensitivity-based assignments such as HAWQ [3] or Q-BERT [4].

3. The experiments rely on BERTBASE with SQuAD and GLUE, models that no longer reflect the current state of quantization research or hardware trends. Modern compression benchmarks typically evaluate on larger or more diverse architectures (e.g., RoBERTa, LLaMA, Qwen). Evaluating only on BERTBASE limits generality and makes the reported gains less impactful.

4. There is no analysis or ablation explaining why mixed-precision across activations and embeddings helps. No sensitivity study, gradient statistics, or visualization of learned bit-width distributions (beyond histograms) is provided to substantiate the observed improvements.

5. Accuracy improvements over existing mixed-precision baselines are relatively small. The practical savings (1.5× memory vs 8-bit baselines) are incremental, especially given the heavy training cost of QAT + KD.

6. The work compares primarily against Q-BERT [4], AQ-BERT [5], and Ternary/Binary BERT, missing recent optimization-based or hardware-aware methods such as SDQ [6], OMPQ [7], or orthogonal mixed-precision frameworks used in contemporary transformers.

References

[1] L. Yang & Q. Jin. FracBits: Mixed Precision Quantization via Fractional Bit-Widths. AAAI 2021.

[2] M. Nikolić et al. BitPruning: Learning Bitlengths for Aggressive and Accurate Quantization. arXiv 2020.

[3] Z. Dong et al. HAWQ: Hessian-Aware Quantization with Mixed Precision. ICCV 2019.

[4] S. Shen et al. Q-BERT: Hessian-Based Ultra-Low-Precision Quantization of BERT. AAAI 2020.

[5] C. Zhao et al. Automatic Mixed-Precision Quantization Search of BERT. arXiv 2021.

[6] X. Huang et al. SDQ: Stochastic Differentiable Quantization with Mixed Precision. ICML 2022.

[7] Y. Ma et al. OMPQ: Orthogonal Mixed-Precision Quantization. arXiv 2021.

**Questions:**

1. What prevents AdaQAT from diverging when applied jointly to activations and embeddings? Have you observed instability in fractional bit-width updates?

2. Can the authors justify the choice of the MSE criterion for layer selection? Did other metrics (e.g., Hessian-based) yield similar behavior?

3. How sensitive are the results to the regularization coefficient or the oscillation threshold?

4. Could the method generalize to larger or instruction-tuned transformers (e.g., LLaMA or RoBERTa), or is it tied to BERT-style architectures?

5. Have you benchmarked actual inference latency on hardware, not just BitOPs estimates, to verify real speed gains?

---

### Official Review · Reviewer_5MRs · 2025-11-01

**Soundness:** 2
**Presentation:** 2
**Contribution:** 1
**Rating:** 2
**Confidence:** 3

**Summary:**

This paper applies existing AdaQAT method to quantize BERT model, covering weights, activations, and embedding layers. The authors integrate layer-wise mixed precision and knowledge distillation into the framework and evaluate on GLUE and SQuAD.

**Strengths:**

It considers quantization for weight, activation, and embedding layers, rather than only weight quantization.

**Weaknesses:**

The novelty of the method is unclear. AdaQAT has been published previously, and the extensions to per-layer optimization and adding distillation appear to be straightforward incremental improvements without new algorithmic techniques.

The baseline comparison does not fully establish superiority. In Table 2 the method compares to BinaryBERT and TernaryBERT, which have different precision targets. Though AdaQAT has better accuracy than BinaryBert, BinaryBERT has significantly lower memory footprint and BitOPs cost, making the comparison inconclusive in terms of efficiency versus quality trade-off. A more comprehensive comparison against methods targeting similar bit configurations would improve fairness and clarity.

More recent quantization approaches are missing from the evaluation, such as QuaRot and other mixed-precision or QAT/PTQ transformer quantization techniques. The method should also report results across a range of bit-width settings to better demonstrate flexibility and the quality-efficiency trade-offs.

The experimental scope is limited. The evaluation is restricted to BERT on GLUE and SQuAD, which are foundational but now considered relatively simple and saturated benchmarks. To support broader impact claims, it would be important to evaluate on more modern language models such as RoBERTa, DeBERTa, or other transformer encoders, and on more challenging benchmarks including retrieval tasks, long-context understanding, code modeling, or multilingual QA. These settings would better reflect current deployment-relevant scenarios and demonstrate robustness of the proposed method.

**Questions:**

Could you analyze and demonstrate your method on more model architectures beyond BERT, for example RoBERTa, DeBERTa, or LLaMA-based encoders?

Could you evaluate on modern and more challenging datasets beyond GLUE and SQuAD, such as multilingual or retrieval benchmarks?

---

### Official Review · Reviewer_WSap · 2025-11-06

**Soundness:** 2
**Presentation:** 2
**Contribution:** 2
**Rating:** 2
**Confidence:** 5

**Summary:**

The paper’s core idea—mixed-precision quantization for BERT (weights, activations, and embeddings)—was an active topic around 2019–2021, but has since been superseded by post-training quantization (PTQ) and hardware-aware inference optimization methods. The community’s focus has decisively shifted toward scalable low-bit inference for large language models (LLMs) and hardware-co-designed quantization (e.g., FP8, FP4, mixed-KV cache quantization), while this work remains confined to fine-tuning BERT on GLUE/SQuAD.

**Strengths:**

The paper presents a systematic exploration of mixed-precision quantization for BERT, covering weights, activations, and embeddings, and reports empirical results on SQuAD and GLUE benchmarks. The overall presentation is clean, and the authors attempt to integrate knowledge distillation to mitigate quantization degradation. The topic—efficient model compression—is broadly relevant to the community.

**Weaknesses:**

This work feels substantially outdated both in scope and technical contribution. The core idea of adaptive mixed-precision quantization for Transformer models was well-studied in 2019–2021 (e.g., Q-BERT, HAQ, AdaQuant, and related works). The paper does not offer any novel algorithmic insight or quantization formulation beyond existing methods; rather, it re-implements known approaches with minor variations.


The experimental setup is limited to BERT-base and legacy benchmarks (GLUE, SQuAD), which are already saturated and no longer representative of current research directions. No results are provided on modern architectures (e.g., decoder-only LLMs) or larger-scale models.

Even for the BERT model, the final weight activation bits shown in Appendix 3 seems to tell us that uniform w2a4 is pretty much good if we train longer and search more better hyper parameters. There seems less value in the  BitOPs, saving extra 10-20MB is not too helpful given the modern hardware.

**Questions:**

The learning rate is one of the most critical hyperparameters in Quantization-Aware Training (QAT). However, the authors only experiment with a single value (2e-5), which is quite limited and raises concerns about whether the reported results are robust to different training configurations.

---

### Note · Authors · 2025-11-25

**Comment:**

We sincerely thank the reviewers for their thoughtful feedback and for the time invested in evaluating our submission. Based on the comments, we recognize that the positioning of our work, specifically, its focus on custom hardware–accelerated inference rather than GPU/TPU FP4/FP6 pathways, was not communicated with sufficient clarity. We also acknowledge that we could have better articulated the improvements introduced in relation to the original AdaQAT paper. While we respectfully disagree with the assessment that PTQ approaches supersede QAT, particularly given that QAT remains impractical for many large-scale LMs, we agree that including comparisons with representative PTQ baselines would strengthen the presentation, and we plan to incorporate such evaluations in future iterations.
Furthermore, we politely disagree with the assertion that BERT has been “shelved” or that comparisons with very large models such as GPT-4 or LLaMA are appropriate for a study not intended for that scale, especially when considering the environmental and computational sustainability of research practices. To address concerns about breadth, we will extend our evaluations to additional mid-size models, including RoBERTa and DeBERTa, and incorporate the Holmes benchmark to provide a more comprehensive assessment of downstream NLP tasks.

Given these considerations, and in order to revise the work thoroughly, we respectfully request to withdraw the submission at this time. We appreciate the reviewers’ comments and will use them to substantially improve the clarity and scope of our future work.

**Withdrawal Confirmation:**

I have read and agree with the venue's withdrawal policy on behalf of myself and my co-authors.